# What Is the Metabolic Amplification of Insulin Secretion and Is it (Still) Relevant?

**DOI:** 10.3390/metabo11060355

**Published:** 2021-06-02

**Authors:** Ingo Rustenbeck, Torben Schulze, Mai Morsi, Mohammed Alshafei, Uwe Panten

**Affiliations:** 1Institute of Pharmacology, Toxicology and Clinical Pharmacy, Technische Universität Braunschweig, D38106 Braunschweig, Germany; to.schulze@tu-bs.de (T.S.); m.morsi@tu-bs.de (M.M.); mohammed.alshafei@tu-bs.de (M.A.); u.panten@tu-bs.de (U.P.); 2Department of Pharmacology, Faculty of Pharmacy, Assiut University, Assiut 71526, Egypt

**Keywords:** cytosolic calcium concentration, glucose, insulin secretion, metabolic amplification, mitochondria, nutrient secretagogues

## Abstract

The pancreatic beta-cell transduces the availability of nutrients into the secretion of insulin. While this process is extensively modified by hormones and neurotransmitters, it is the availability of nutrients, above all glucose, which sets the process of insulin synthesis and secretion in motion. The central role of the mitochondria in this process was identified decades ago, but how changes in mitochondrial activity are coupled to the exocytosis of insulin granules is still incompletely understood. The identification of ATP-sensitive K^+^-channels provided the link between the level of adenine nucleotides and the electrical activity of the beta cell, but the depolarization-induced Ca^2+^-influx into the beta cells, although necessary for stimulated secretion, is not sufficient to generate the secretion pattern as produced by glucose and other nutrient secretagogues. The metabolic amplification of insulin secretion is thus the sequence of events that enables the secretory response to a nutrient secretagogue to exceed the secretory response to a purely depolarizing stimulus and is thus of prime importance. Since the cataplerotic export of mitochondrial metabolites is involved in this signaling, an orienting overview on the topic of nutrient secretagogues beyond glucose is included. Their judicious use may help to define better the nature of the signals and their mechanism of action.

## 1. The Biphasic Pattern of Insulin Secretion Can Be Produced in Different Experimental Settings

The endocrine pancreas responds to an increase in glucose in the bloodstream with a biphasic pattern of insulin release, represented by a short (5–10 min) first phase, followed by a long, sustained second phase release. This unique feature is best shown in response to a ‘square wave’ glucose stimulus, which is produced in vivo by the hyperglycaemic clamp [1]. It is obvious that such a stimulation pattern is non-physiological, but the resulting biphasic response is the hallmark of the healthy endocrine pancreas. In type 2 diabetes (T2D) and animal models of this disease, it displays varying degrees of a diminished insulin response, often described to be more prominent during the first phase but also recognizable during the second phase [2,3]. The demonstration that metabolically healthy first-degree relatives of T2D patients have a diminished secretory response during a hyperglycaemic clamp [4] and that insulin-sensitive offspring of T2D patients have diminished beta-cell function [5] suggest that beta-cell dysfunction is an independent pathogenic factor driving the progressive impairment of glucose homeostasis in the course of T2D [6]. The low incidence of T2D during historical periods of food shortage shows that beta-cell dysfunction is rarely sufficient for manifestation in the absence of insulin resistance.

Originally demonstrated in vivo, the biphasic secretion pattern could also be demonstrated in vitro, or more precisely, ex vivo, using the model of the perfused rat [7,8] or mouse pancreas [9] and, after the introduction of the collagenase isolation technique, with the more reductionist model of the batch-perifused isolated islets [10,11]. Theoretically, it is possible that the phenomenologically similar secretion pattern in vivo and ex vivo may result from different mechanisms. In a less apodictic form, namely that the relative contribution of the processes involved in shaping the secretion pattern may vary between the different experimental systems, this argument is worth considering (see below). Just to illustrate this point, the effect of incretins is certainly absent in perifused islets, which may be responsible for their more protracted and less dynamic responses.

Today, a large part of the basic research on beta-cell function is performed using insulin-secreting cell lines, often referred to as clonal beta cells. In addition to the initially perceived advantage, to generate a steady supply of insulin-secreting cells with standardized properties without the hazards and time consumption of collagenase digestion, the ease of transfection has made them popular tools. Thus, cell lines are also used for the study of metabolism-derived signals for secretion (see, e.g., [12,13]). However, it has to be kept in mind that the much larger biosynthetic activity as a precondition for the higher rates of mitosis represents a drain of metabolites. This is one of the reasons why quickly proliferating cell lines, such as RINm5F, have a poor secretory response to metabolic stimulation, whereas highly differentiated cell lines, such as INS-1E, have a better response but proliferate only slowly [14,15]. Interestingly, three-dimensional aggregates of MIN6 cells, also called MIN6 pseudo-islets, show a higher secretory response to glucose than the equivalent number of MIN6 cells in monolayer [16] and can be perifused like primary islets. Still, the kinetics and dynamic range of secretion are inferior to the one of primary islets.

## 2. The Early Years: Substrate Site Hypothesis versus Receptor Site Hypothesis 

Very early investigations, using a number of metabolizable and non-metabolizable carbohydrates, suggested that stimulated insulin secretion required the metabolic breakdown of the glucose [17,18]. For example, glucose and mannose stimulated insulin secretion, whereas 2-deoxyglucose, 3-*O*-methylglucose, and galactose did not. However, it was difficult to verify that increases in the level of potential signal-conveying metabolites occurred during the initial phase of stimulated secretion; thus, the alternative view evolved, namely that glucose stimulates insulin secretion by binding to a plasma membrane receptor [19]. The demonstration that the NAD(P)H-autofluorescence of perifused islets increased, coinciding with the onset of stimulated secretion, suggested that the level of mitochondrial reducing equivalents increased during the onset of secretion [20] and supported the substrate site hypothesis.

Perhaps, some of the earlier controversies were caused by the existence of sweet taste receptors at the beta-cell plasma membrane, the activation of which may lead to a leftward shift of the threshold of glucose stimulation [21]. The receptor concept survived in modified form when the prominent role of the glucokinase, the first glucose-metabolizing enzyme in the beta cells, to regulate the velocity of glucose breakdown was recognized [22,23,24]. Actually, the biochemical or pharmacological enhancement of the glucokinase activity proved to have insulinotropic consequences [25,26]. Nevertheless, it remained unclear for some time whether proximal events of glucose metabolism could directly affect insulin secretion [27] or whether the complete degradation of the glucose carbon to CO_2_ and increased oxidative phosphorylation was necessary for signal generation and not just a complicating by-product of glucose metabolism.

## 3. What Are Nutrient Secretagogues?

The metabolization of glucose as a precondition for its insulinotropic effect has become textbook knowledge. In consequence, not only the earlier disputes but also, the earlier experimental approaches to characterize the relation between the nutrient and the insulinotropic properties are becoming less widely known. In view of the still unresolved issues, they may not only be of historical interest. The insulinotropic property of non-carbohydrates belongs to this chapter.

In addition to the few insulinotropic carbohydrates, some amino acids and monocarboxylic keto acids are insulinotropic. However, the concentrations needed to elicit insulin secretion are beyond their physiological ranges in the serum of humans and mice. So, they cannot be named physiological stimulators, even if the mechanisms they activate within the beta cell are physiological in the sense that even prolonged stimulation is not detrimental for beta-cell function and viability. This sets them apart from more robust stimuli, such as depolarization, by high extracellular potassium concentration (see below). The feature which made (and still makes) amino acids and keto acids interesting as heuristics tools is that their catabolism and the one of glucose only meet in the mitochondrial matrix, thus reducing the number of possible mechanisms involved in stimulus-secretion coupling.

The amino acid which is able to elicit insulin secretion in the absence of glucose is leucine. Its insulinotropic efficacy is strongly enhanced by combination with glutamine, which by itself is ineffective [28,29]. The series of keto acids are of interest since their catabolism shows differences, as shown by their insulinotropic efficacy. Alpha-ketoisocaproic acid (KIC), alpha-ketocaproate, alpha-ketovalerate, and alpha-keto-beta-methylvalerate are all insulinotropic [30]. KIC has the strongest insulinotropic effect, on a par with glucose, albeit with different kinetics. Alpha-ketoisovalerianic acid (KIV) is not insulinotropic but can enhance the efficacy of other nutrient secretagogues. Actually, the term nutrient secretagogue for such compounds is preferable over “fuels” since they do not only support ATP generation but like glucose, support the biosynthesis of insulin, as has been shown for the combination of leucine and glutamine [31].

All of the above insulinotropic compounds support the generation of reducing equivalents by the Krebs cycle in the mitochondrial matrix, either by being catabolized themselves or by activating the catabolism of endogenous nutrients. Like leucine, BCH (2-amino-bicyclo [2.2.1]heptane-2-carboxylic acid), a non-metabolizable structural analogue of leucine, activates glutamate dehydrogenase and provides the citric acid cycle with alpha-ketoglutarate [32]. Similarly, beta-phenylpyruvate is not metabolizable but serves as transamination partners for glutamate and glutamine [33]. In the course of phenylketonuria, an inborn error of metabolism, phenylpyruvate can reach concentrations that inappropriately stimulate insulin secretion and generate hypoglycaemia [34]. The role as a transamination partner is also relevant for KIC, which is itself metabolized, yielding acetyl-CoA, but also transaminates glutamate and thus provides the Krebs cycle with alpha-ketoglutarate [30]. Inhibition of transamination by aminooxyacetate abolishes the insulinotropic effect of KIC [35].

However, it has to be emphasized that in contrast to glucose, not all of the keto acids are pure nutrient secretagogues. For example, KIC and beta-phenylpyruvate were found to directly inhibit the ATP-sensitive K^+^ channel (K_ATP_ channel) [36,37] in addition to inhibiting the channel by the increased ATP/ADP ratio [38,39]. This has to be kept in mind when interpreting secretory responses to keto acid stimulation. In this context, it should be mentioned that distinct from other amino acids, the insulinotropic property of arginine is not caused by metabolism but is exclusively due to the depolarization caused by its electrogenic uptake into the beta cells [40].

## 4. Which Features of the Mitochondrial Metabolism Are Typical for Beta Cells?

The insulinotropic property of the above amino- and keto-acids play a major role in defining the mitochondria as the central hub of the stimulation of insulin secretion by nutrients (see Figure 1). Several lines of evidence contributed to this progress. First, the demonstration that not only glucose but nutrient secretagogues in general increase NAD(P)H-fluorescence and at the same time decrease FAD-fluorescence [41,42,43], which points to the mitochondrial matrix. Second, KIC, like glucose and a number of other nutrient secretagogues, increases oxygen consumption supporting a role for oxidative phosphorylation [44,45,46]. By combining oxygen consumption measurements of perifused islets with secretion measurements, an interesting difference in the kinetics was observed, in that the increase in KIC-stimulated secretion closely parallels the increase in the oxygen consumption rate (OCR), whereas glucose generates a faster increase in the OCR, which then turns into a slowly ascending plateau [47].

It is interesting that islet cells in situ live in a higher oxygen tension than the surrounding exocrine tissue [48], even though mitochondria in beta cells make up only 4% of the intracellular volume, much less than, e.g., in hepatocytes [49]. The higher oxygen tension is likely a consequence of dense vascularization of the islet [50,51]. It is known that three critical dehydrogenases of the Krebs cycle can be activated by increased Ca^2+^ levels [52]. Interestingly, the increase in oxygen consumption, which results from this activation, occurs only when Ca^2+^ is provided by influx via L-type Ca^2+^ channels, not by release from intracellular stores [53].

The role of mitochondrial metabolism for glucose-derived signal generation is emphasized by the fact that the fraction of glucose carbon that is oxidized increases with accelerated glycolysis, even though it is already high at basal levels. Another notable feature is that glucose utilization does not increase with decreasing oxygen levels [54]. The latter feature is explained by the low activity of lactate dehydrogenase in beta cells [55]. Thus, to re-oxidize NADH in the cytosol and to keep the glycolysis running, reducing equivalents have to be transported into the mitochondrial matrix via shuttling mechanisms. This is enabled by a high expression level of the FAD-linked glycerol-3-phosphate dehydrogenase [56]. The use of pyruvate in beta cell mitochondria is characterized by a high activity of pyruvate carboxylase, which is atypical for a cell without gluconeogenic function [57].

Finally, the specific relevance for the beta cell of the close coupling of glucose utilization to mitochondrial metabolism is underlined by the much looser coupling in the glucagon-secreting alpha cells [54]. Given the central position of the mitochondria in the stimulus-secretion coupling, it is not surprising that they are considered as essential contributors to the pathogenesis of type 2 diabetes [58].

## 5. Which Observations Have Led to the Hypothesis of a Bifurcating Pathway Emanating from the Mitochondria?

The realization that nutrient-stimulated insulin secretion is dependent on mitochondrial activation, or more precisely, the increased oxidative phosphorylation, led to the hypothesis that ATP or perhaps the ATP/ADP-ratio might convey the signal to the electrical activity which had been observed in beta cells [59,60]. Measurements of radioactivity from perifused islets loaded with rubidium (^86^Rb) identified a decrease in the potassium conductance as the likely initiating event of electrical activity, which was required for the initiation of exocytosis [61]. However, even though it could be shown that raising the glucose concentration increased the ATP content in islets, it did not occur in the insulinotropic concentration range of glucose, and measurements of the ATP/ADP ratio yielded inconclusive results [62,63]. Only when the high background by the ATP content in the insulin granules was diminished by partial degranulation could a convincing correlation between the ATP/ADP ratio and insulin secretion be demonstrated [64].

The missing link between changes in the ATP/ADP ratio and the electrical activity proved to be the K_ATP_ channel, the functional identification of which was a fruit of the newly developed patch clamp technique [65,66]. It took several years to clarify its molecular composition, which consists of an inwardly rectifying K^+^ channel (Kir 6.2 = KCNJ11) as the pore-forming unit and a member of the ATP cassette-binding family (SUR1 = ABCC8) as the regulatory subunit [67]. While ATP^4-^ closes the channel by interaction with the channel pore, Mg-ADP binding to the regulatory subunit (also termed sulfonylurea receptor) has an opening effect on the channel and modifies sulfonylurea potency [68,69]. Even though additional metabolism-derived signals, such as H_2_O_2_ or acyl-CoA, may contribute to the regulation of K_ATP_ channels [70,71], the ATP/ADP ratio as determined in intact islets can be considered as a sufficient measure of the signal by which the beta cell mitochondria initiate the electrical activity of the plasma membrane [72].

After the identification of the K_ATP_ channel as the link between energy metabolism and electrical activity of the beta cell, the view prevailed that the depolarization induced Ca^2+^ influx caused by the closure of the channel would constitute the final common pathway of insulin secretion. This view was regarded as the consensus theory of glucose-induced insulin secretion. A test of this theory was to close the K_ATP_ channel pharmacologically and then raise glucose from a basal to a stimulatory level. If Ca^2+^ influx by K_ATP_ channel closure was the only regulator of secretion, no further increase in secretion was to be expected. However, glucose caused an increase in secretion beyond the level established by the maximally effective concentration of sulfonylurea [73]. The consensus theory was further challenged by another experimental approach, where K_ATP_ channels were opened by diazoxide to prohibit any effect of the energy metabolism on the membrane potential. Depolarization was induced by a high extracellular potassium concentration. Because of the large potassium conductance under this condition, the membrane potential closely follows the Nernst potential. Again, raising glucose to a stimulatory level increased the secretion of isolated islets beyond the level established by depolarization alone [74].

The latter approach has become the experimental standard, in part because of suspected insulinotropic effects of sulfonylureas in addition to the closure of plasma membrane K_ATP_ channels [75,76]. However, interference by intracellular sites of action also limits the value of diazoxide as a pharmacological tool since it was shown to exert direct effects on beta-cell mitochondria [77]. An argument in favor of pharmacologically blocking the K_ATP_ channels is that the depolarization thus produced corresponds to the depolarization strength and pattern of nutrient stimulation, whereas depolarization by high potassium can be much stronger, depending on the actual concentration, and lacks the typical pattern of action potential spiking [78,79].

Initially often named “K_ATP_ channel-independent pathway” [80], the name “amplifying pathway”, as suggested by Henquin [81], has become widely accepted. This choice of name was based on the evidence that insulin secretion is not increased if the plasma membrane is not depolarized, which under physiological conditions requires the closure of the K_ATP_ channels. For this reason, the pathway is not “independent” in the proper sense. The pathway leading to the K_ATP_ channel closure and Ca^2+^ influx, in contrast, was termed “triggering pathway” [81]. Another nomenclature tries to avoid a functional definition by simply naming the stimulus-secretion coupling via K_ATP_ channel closure the “canonical pathway” and leaving the role of those signals undefined, which affect secretion without causing changes in the electrical activity [82]. What is important, though, is not to confuse the metabolic amplification, an inherent property of nutrient secretagogues, with a receptor-mediated enhancement of secretion by neurotransmitters or hormones, such as GLP-1.

## 6. Models of the Biphasic Kinetics of Secretion 

A certain weakness in the definition of the amplifying pathway is that it relies on the use of pharmacological agents prior to the nutrient stimulation. This, by necessity, influences the secretion kinetics of the nutrient. So, to discuss whether the metabolic amplification primarily affects early or late periods of the biphasic secretion pattern, we have to briefly touch on the topic of which mechanisms underlie the biphasic kinetics of secretion. In principle, two different points of view can be distinguished: one considers the limited number of secretion-ready granules as the main responsible factor, whereas the other considers the evolving pattern of metabolism-derived signals as the underlying cause.

The more popular hypothesis proposes that the biphasic secretion pattern results from the existence of two different pools of granules. Its origins can be traced back to the two-compartment model of secretion, proposing that insulin is contained in a stable compartment and in a much smaller compartment that is labile to stimulation [83]. Based on the combination of ultrastructural and electrophysiological data, the correlate of the latter compartment was suggested to be a limited number of secretion-ready granules that are firmly attached to the plasma membrane (“docked”) and fully prepared (“primed”) for fusion [84,85,86]. The high fusion rate produced by depolarization-induced Ca^2+^ influx depletes this pool of secretion-ready granules causing the transient decrease in the rate of insulin secretion, observed after 10–15 min of continuous stimulation (somewhat longer than in vivo). The progressively faster replenishment of this pool by metabolism-dependent recruitment of a distant reserve pool causes the subsequent recovery in the insulin release rate, taking shape as the second phase [87,88]. Since the accelerated rate of replenishment now matches the rate of exocytosis, the secretion rate of the second phase can be maintained for several hours until a general desensitization towards stimulation sets in, which reduces the secretion rate to about one-third [89].

The alternative view can be named the metabolic control hypothesis. This hypothesis proposes that the glucose metabolism in the beta cell generates stimulatory as well as inhibitory signals [90,91]. The initial stimulatory effect is followed by a more slowly evolving inhibitory signal which reduces the secretion rate. This process, which leads to the nadir of the secretion rate, was named time-dependent inhibition. The hypothetical inhibitory signal is then succeeded or overruled by a third signal evolving from the glucose metabolism. This even more retarded reaction was named time-dependent potentiation. The time-dependent inhibition and potentiation do not only differ in their kinetics but also their dependency on the glucose concentration [92]. The main obstacle for this hypothesis to gain a wider acceptance is that no clear mechanism for the inhibitory signal could be demonstrated.

On the other hand, the hypothesis of the time-dependent potentiation bears a clear resemblance to the phenomenon of metabolic amplification. A comparison of the storage- and the signal-limited model of insulin secretion gave slightly better fits for the simulation of the signal-limited model [93]. Further progress in characterizing the fate of the insulin granules may eventually show that both hypotheses are not mutually exclusive but rather that the earlier model (limited storage) is related to the latter.

In this context, it is worth pointing out that a fully developed first phase with a transient decrease in the secretion rate requires the “non-physiological” square-wave stimulus (Figure 2). A ramp-like stimulation generates a continuously ascending rate of secretion, where the initial phase is only different from the following phase by a more steeply ascending rate of secretion [94]. In perifused islets, such a pattern also results when a maximally effective glucose stimulus is applied after a period of nutrient deprivation [47,95]. This suggests that the velocity with which metabolites are generated may not only determine the initial increase in the secretion rate but also the extent of the transient decrease.

## 7. Relation of the Triggering and Amplifying Pathways to the Biphasic Secretion Kinetics

The hypothesis that the imbalance between the fusion rate of the fully prepared granules and the velocity of granule replenishment is responsible for the first phase entails that the relevance of the mitochondrial metabolism for secretion becomes visible during the build-up of the second phase. Consequently, the triggering signal was often held to be responsible for the first phase and the amplifying signal for the second phase of glucose-stimulated secretion, see, e.g., [96].

This view was supported by observations on INS1 cells with inactivated mtDNA, which did not secrete insulin in response to glucose but did so in response to KCl depolarization [97]. However, the short-term treatment of perifused islets with mitochondrial inhibitors virtually abolished not only the insulin secretion elicited by glucose but also the one by KCl or sulfonylureas [98]. MIN6 cells devoid of mitochondrial DNA showed a markedly reduced secretion in response to the sulfonylurea, glibenclamide [99]. So, the extent to which the triggering signal alone elicits secretion is less well defined than often realized. In this context, it is worth mentioning that Ca^2+^ influx does not simply constitute the signal for granule fusion but that it may affect granule transport and thus the availability of fusion-ready granules. In fact, the biphasic increase in the cytosolic Ca^2+^ concentration, often seen in intact islets [100,101], suggests a role for the Ca^2+^ signal beyond triggering.

As mentioned above, potassium depolarization has no upper limit within the physiological working range of the beta cell, in contrast to K_ATP_ channel closure. The closure of the K_ATP_ channels results in a plateau depolarization of about 20 mV by a still ill-defined inward leak current. The superimposed action potentials result from the phasic influx of Ca^2+^ via L-type channels, as can be seen by the use of blockers of these channels [102]. So, for the experimental triggering signal to mimic the consequence of K_ATP_ channel closure by nutrient stimulation, a depolarization by about 20 mV has to be produced to open voltage-dependent Ca^2+^ channels. This is the depolarizing strength of 15 mM KCl, which only elicits a small transient increase in secretion far below the typical range of the first phase [102]. Of note, the presence of 15 mM KCl is not without effect. It enhances the secretion elicited by glucose and by sulfonylureas [102,103].

These observations suggest that a strong depolarization (e.g., by 40 mM KCl and more) can produce a first phase-like monophasic secretion, but that the weaker depolarization produced by nutrient stimulation generates the first phase because the metabolic amplifying is already effective. This conclusion concurs with the one drawn from observations on the changing relation between first and second phase secretion in dependence on the prestimulatory conditions [104]. In short, metabolic amplification is likely effective from the beginning of nutrient-stimulated secretion.

Finally, it has to be mentioned that the live-cell imaging of fluorescently labelled granules by TIRF microscopy have led to a much more detailed view on the insulin granules and the events preceding the fusion process, see, e.g., [105,106,107]. However, these data have not yet had a major impact on the hypothesis of triggering and amplifying pathways. Certainly, the quantitative description of granule genesis, transport, and fusion under triggering and amplifying conditions will be of major help to understand how the amplifying signals enhance the secretion rate.

## 8. Cataplerosis and Putative Amplification Signals 

In contrast to the triggering pathway, a broadly accepted model of the amplifying pathway has not yet emerged. A consensus exists in that the export of metabolites from the mitochondrial matrix into the cytosol (cataplerosis) is of importance [108]. Cataplerosis is any process by which carbon leaves the citric acid cycle in a form different from carbon dioxide. Thereby the need for refilling the citric acid cycle (anaplerosis) is generated. So, it can be hypothesized that a compound qualifies for the role of a nutrient secretagogue when it increases the rate of oxidative phosphorylation and at the same time supports cataplerosis by refilling the citric acid cycle [57].

According to this theory, glucose is insulinotropic not only because it generates NADH and FADH_2_ in the matrix space but also because it refills the pool of oxaloacetate by the action of pyruvate carboxylase. Actually, only 60% of the glucose-derived pyruvate entering the mitochondria is metabolized by pyruvate dehydrogenase to acetyl-CoA; the other 40% is built up to oxaloacetate. This compensates for the loss of cataplerotic metabolites and permits the constant running of the citric acid cycle [109]. This scenario would apply when mitochondria release glutamate, which could be generated from alpha-ketoglutarate by the glutamate dehydrogenase reaction. Glutamate was the first in the line of candidate signal-conveying compounds [110], also comprising alpha-ketoglutarate [111] or short-chain acyl-CoA [112].

Some authors view the cataplerosis as part of one or more cycles of export and re-import of metabolites to convey reducing equivalents out of the mitochondria to ultimately increase the cytosolic levels of NADPH [113,114]. NADPH is discussed as a regulator of exocytosis in beta cells via the deSUMOylation of SNARE proteins by SENP1 protein [115,116]. It has been shown repeatedly that glucose stimulation increases the level of NADPH (not to be confused with NAD(P)H, which designates the sum of both NADH and NADPH in conventional fluorometry). But such an increase was not found for the amplifying effect of KIC [117]. Thus, NADPH is unlikely to be an indispensable signal for metabolic amplification. From a wider perspective, it can be postulated that candidate signalling compounds, such as monoacylglycerol [118], which can result from the metabolism of glucose, but not of nutrient secretagogues in general, may contribute to the metabolic amplification but are unlikely to be indispensable for this process.

As mentioned above, KIC has two different sites of entry into the citric acid cycle; in addition to acetyl-CoA, it can generate alpha-ketoglutarate as long as the transamination partner glutamate is available. The comparison of the metabolic amplification by KIC and by glucose has led to an unexpected result. To obtain a metabolic situation of the beta cells, where KIC metabolism was only minimally affected by the preceding glucose metabolism, a 60 min nutrient-free perifusion preceded the application of exogenous nutrient. The triggering signal was provided by blocking the K_ATP_ channels with a maximally effective concentration of sulfonylurea. The insulinotropic effect of KIC under this condition was about as strong as in the absence of the sulfonylurea (KIC at ˃5 mmol/L is a K_ATP_ channel blocker in its own right), whereas the insulinotropic effect of glucose under the same condition was virtually abolished and did not recover within the next hour. Or, in other words, the metabolic amplification of glucose, but not by KIC, had been blocked by the exposure to sulfonylurea throughout the experiment [119]. Later, it was confirmed that the moderate depolarization by 15 mM KCl could substitute for the action of the sulfonylurea [47].

It can thus be hypothesized that prolonged depolarization in the absence of nutrients leads to a critically low concentration of a metabolite that cannot be replenished by glucose alone. This view was supported by the observation that the secretion can be restarted by adding KIV [120] or by glutamine, the effect of which could be enhanced by BCH [47]. Remarkably, the insulinotropic effect of glucose, once restarted, remained elevated after removal of the anaplerotic starters, demonstrating that the regained rate of anaplerosis translates into the rate of secretion (Figure 3 and Figure 4). Taken together, the above findings and considerations are consistent with the view that mitochondrial export of citrate and acetoacetate, both sources of cytosolic acetyl-CoA, in insulin-secreting cells, amplifies insulin secretion by enhancing the cytosolic supply of acetyl-CoA [120,121,122].

Recently, evidence on the role of cAMP as a signal-conveying compound of glucose-stimulated insulin secretion has been presented [123], reflecting the earlier suggestion of cAMP as the mediator of the time-dependent potentiation [124]. While the enhancing function of receptor-mediated cAMP increase on insulin granule exocytosis is beyond doubt [125], it is difficult to relate cAMP generation to the cataplerotic export of citrate cycle metabolites. As a consequence, this hypothesis reduces the role of cataplerosis in stimulus-secretion coupling to maintaining the biosynthesis of insulin in response to nutrient stimulation. A more complete view of how nutrient-stimulated granule biogenesis, transport, and degradation contribute to secretion kinetics is needed to consider this possibility.

## 9. Concluding Remarks

In this review, we have tried to outline the evolution of the ideas on how nutrient secretagogues stimulate insulin secretion. The central position of the mitochondrial metabolism in this process can be regarded as firmly established. Its specific features in the beta-cell suggest that cataplerosis is a major contributor to nutrient-induced signaling and likely produces the phenomenon of metabolic amplification. A broad consensus on the specific nature of the signaling compound(s) and on its (their) mechanism(s) of action has so far failed to emerge. For a review dealing specifically with the shortcomings of current models of stimulus-secretion coupling, the reader is referred to [126]. Here, we have placed an emphasis on the alternative experimental protocol and on the use of amino acids and keto acids as experimental tools to delineate the mechanisms operative during the metabolic amplification (for an encompassing overview, see [127]). These observations, which were obtained with freshly isolated islets, support the view that in addition to the ATP/ADP ratio, the second branch of signaling originates from the beta-cell mitochondria. Whatever its precise nature, it will impact the provision of granules to the sites of exocytosis. It is conceivable that multiple sites of action are involved in this process, which makes the concept of reversible lysine acetylation by increased levels of cytosolic acetyl-CoA attractive [128]. Based on the modeling of clinical and experimental data, defective metabolic amplification has been suggested to underlie the beta-cell dysfunction during type 2 diabetes [129]. Thus, metabolic amplification continues to be a relevant topic in diabetes research, and its mechanisms warrant further investigation.

## Figures and Tables

**Figure 1 metabolites-11-00355-f001:**
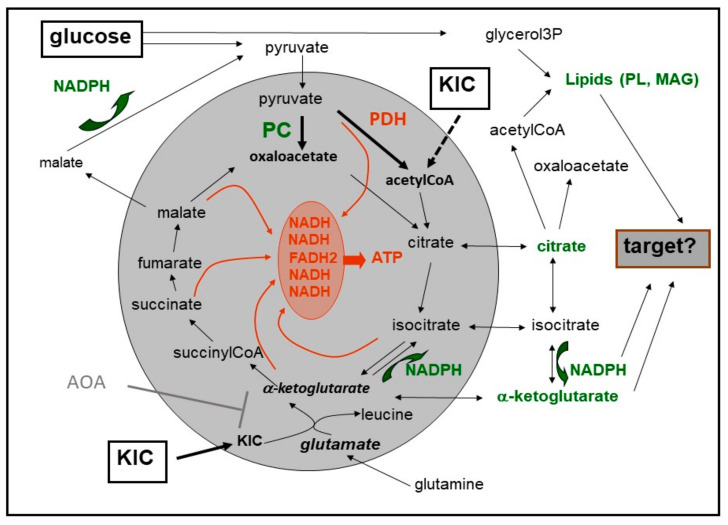
Both glucose and KIC exert an anaplerotic effect on the citric acid cycle in beta cells. Glucose-derived pyruvate is either metabolized by pyruvate dehydrogenase (PDH) to acetyl-CoA or by pyruvate carboxylase (PC) to oxaloacetate. KIC can be degraded, which yields acetyl-CoA. It can also be transaminated with glutamate, which yields alpha-ketoglutarate and leucine. This reaction is essential for the insulinotropic effect since inhibition of transamination by AOA inhibits KIC-induced insulin secretion. The carbon which is fed into the citric acid cycle is derived from glutamate, which, in turn, is generated from endogenous glutamine. The red color marks reactions which support oxidative phosphorylation, the green color marks metabolites and reactions with potential relevance for glucose-dependent amplification. Abbreviations: AOA = aminooxyacetate, KIC = alpha-ketoisocaproic acid, MAG = monoacylglycerol, PL = phospholipids.

**Figure 2 metabolites-11-00355-f002:**
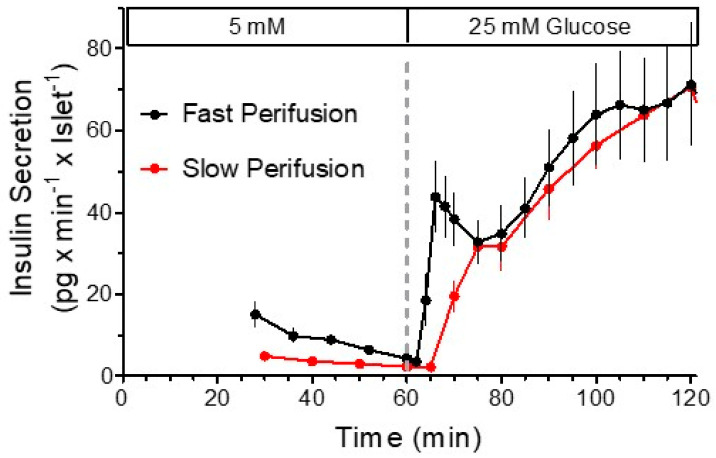
The kinetics of nutrient-induced insulin secretion are strongly dependent on the velocity of nutrient increase. The only difference between the experimental conditions of the secretory responses shown in this figure is the pump rate of the perifusion and, by consequence, the steepness of the glucose gradient. The final concentration was reached after 5 min or 35 min, respectively.

**Figure 3 metabolites-11-00355-f003:**
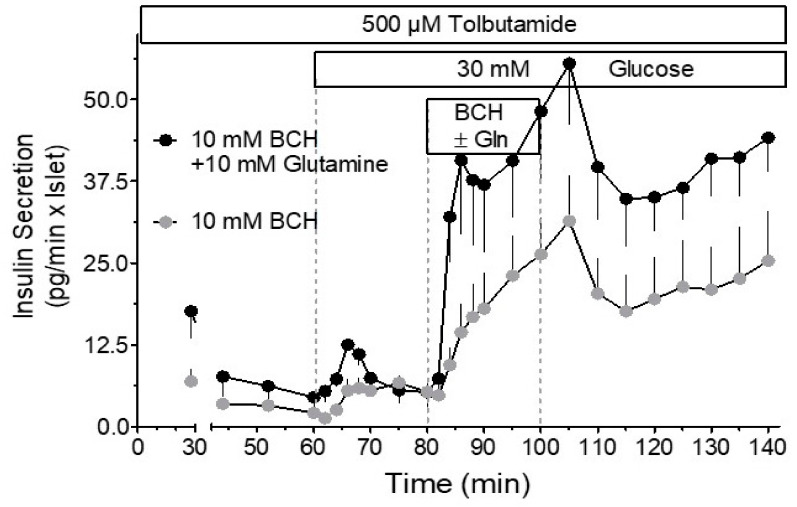
The rescue of the metabolic amplification of glucose, abolished by a period of depolarization in the absence of nutrients. The amplification was restarted by perifusion with BCH and continued after wash-out of BCH. The concomitant presence of glutamine and BCH resulted in a virtual jump-start. Again, the elevated secretion level continued after wash-out of the starters. Since tolbutamide depolarizes beta cells even in the absence of glucose, the triggering signal is continuously present. For the underlying mechanisms, see Figure 4. Adapted from Reference [47].

**Figure 4 metabolites-11-00355-f004:**
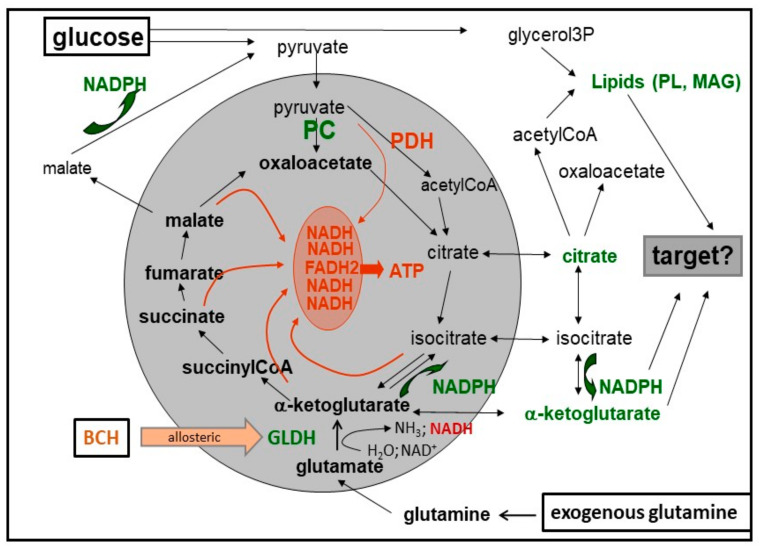
The provision of the citric acid cycle with alpha-ketoglutarate may not only run via transamination but also via glutamate dehydrogenase. This enzyme is allosterically activated by leucine and by (±) BCH, a non-metabolizable leucine analog. In principle, the reaction can run in either direction, but normally the level of endogenous glutamate is sufficient to support the generation of alpha-ketoglutarate. The deaminating, anaplerotic reaction is further enhanced by the exogenous offer of glutamine. The cataplerotic export of acetoacetate, which depends on sufficient levels of succinate, is omitted for clarity. Abbreviations: BCH = 2-amino-bicyclo [2.2.1]heptane-2-carboxylic acid, GlDH = glutamate dehydrogenase.

## Data Availability

The figures are original work of the authors.

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
