# Peer review of "What Is the Metabolic Amplification of Insulin Secretion and Is It (Still) Relevant?"

_metabolites, 2021, doi:10.3390/metabo11060355_

Round 1
Reviewer 1 Report
This is an excellent review of the role of metabolism in insulin secretion. Though the main goal is to describe the present state of uncertainty about the amplifying pathway of secretion in section 8, the discussion ramps up slowly to that set of questions by recapitulating the long and winding historical path to establishing that glucose metabolism, and not receptor binding, govern secretion. The authors make a good point at the beginning of section 3 that the early disputes and experimental approaches that built the foundation of current understanding are being lost.
I have only a few minor substantive comments and grammatical suggestions.
Conceptual comments:
l. 37: first phase vs second phase role in T2D: Ref. # 127 could be cited here as well as #2 and #3.
l. 40: "beta cell dysfunction is an independent pathogenic factor": it might be worth adding that, though pre-existing, it does not become manifest in the absence of insulin resistance, except in extreme cases.
Section 6, first paragraph: granule pools are counterposed to metabolism generated signals: In my opinion, the metabolism generated signals exert their effect by enhancing granule trafficking and modification, so these are not really exclusive explanations.
English usage comments:
l. 12: "has been identified" would be better as "was identified"
l. 13: "the way how changes" should be "how changes"
l. 57: "easy transfection" would be better as "ease of transfection"
l. 110: "as shows their insulinotropic efficacy" should be "as shown by ..."
l. 198: "Only when ... a convincing correlation ... could be demonstrated" should be "Only when ... could a convincing correlation ... be demonstrated"
l. 406: "to maintain" should be "to maintaining"
Reviewer 2 Report
In the article titled What is the metabolic amplification of insulin secretion and is it (still) relevant authors described the metabolic amplification of insulin secretion in pancreatic beta cells. This is an interesting review of the often neglected amplifying pathway. Below I have listed a few of my inquiries and comments for the article.
- Line 51: is instead of “it”
- Lines 53-66: Although the authors described some differences between islets and cell lines used for research, it should be emphasized that cell lines have some significant drawbacks. They do not only have an inadequate metabolic response, but they also do not respond to physiological glucose concentrations but respond only to a much higher glucose load.
- Line 120: BCH abbreviation is not defined
- Figure 1 caption: all abbreviations presented in figure 1 should be described in the figure caption
- Line 167-168: Authors described the differences in oxygen consumption during Ca2+ influx and release from the intracellular stores. Could you comment or speculate on the possible mechanism for such differentiation of Ca2+ source?
- Line 201: abbreviation of the ATP-sensitive K+ channel is missing
- Line 220: of sulfonylurea instead of “a sulfonylurea”
- In sections 6 and 7 authors described different models of the biphasic kinetics of secretion and their relation. Both described models are based on the properties of individual beta cells. How intrinsic heterogeneity of cells contribute to the observed pattern of secretion? It has been shown that beta cells are coupled to a functional syncytium. Could you comment or speculate how collective activity could contribute to the dynamics of secretion?
- Lastly, could some of the metabolites exported from the mitochondrial matrix serve as intercellular messengers acting either via a paracrine mechanism or passing Cx36 and thereby allowing metabolic communication between cells and their synchronous response to stimuli?
Reviewer 3 Report
Rustenbeck et al discuss the transduction of the nutrient availability to beta cell insulin release code. Currently is this connection predominantly, although notably subadequately, explained in a way that beta cell metabolism, or more precisely mitochondrial ATP production links to ATP-dependent plasma membrane proteins to control the plasma membrane polarization and Ca2+ influx leading to insulin release. To compensate the missing fraction of the insulin release, the metabolic amplification, including cataplerotic export of mitochondrial metabolites has been called for help. The major strength of the paper is that it deals with a nutrient metabolism, which is rarely discussed at this high detail in the literature. However, there are some conceptual and mutually interconnected issues that prevent a move to a more complete understanding of the biphasic insulin release in beta cells. First glucose has been rarely used in physiological concentration range, pharmacological stimulation happens at saturating concentrations, not to mention the KCl as completely unphysiological stimulation. Second, in explaining the biphasic insulin release, authors fail to mention that already the cytosolic Ca2+ profile, especially in measured in fresh pancreatic tissue slices stimulated with a physiological level of glucose is per se biphasic and could as well contribute to secretory vesicle mobilization as well as the metabolic control. In their search to support either substrate site or receptor site hypothesis, the authors could have reached deeper. At least to the level of intracellular membranes and Ca2+ release channels, that are also responsive to ATP and cAMP. Without addressing these open issues also decoding of metabolic control of insulin release will stay incomplete.
In the chapter 8, the references to Figures 3 and 4 are lacking in the text. Including a paragraph or two to explain these would surely contribute to the clarity of the review.
In Figure 2 it is not clear for the different perifusion regimes at which point the final glucose concentration has been reached.
